# Instability of explicit time integration for strongly quenched dynamics with neural quantum states

**Hrvoje Vrcan**[1*] **and Johan H. Mentink**[1†]

**1** Radboud University, Institute of Molecules and Materials,
Heyendaalseweg 135, 6525AJ Nijmegen, The Netherlands

* hrvoje.vrcan@ru.nl , † johan.mentink@science.ru.nl

## Abstract

Neural quantum states have recently demonstrated significant potential for simulating quantum dynamics beyond the capabilities of existing variational ansätze. However, studying strongly driven quantum dynamics with neural networks has proven challenging so far. Here, we focus on assessing several sources of numerical instabilities that can appear in the simulation of quantum dynamics based on the time-dependent variational principle (TDVP) with the computationally efficient explicit time integration scheme. Using the restricted Boltzmann machine architecture, we compare solutions obtained by TDVP with analytical solutions and implicit methods as a function of the quench strength. Interestingly, we uncover a quenching strength that leads to a numerical breakdown in the absence of Monte Carlo noise, despite the fact that physical observables don't exhibit irregularities. This breakdown phenomenon appears consistently across several different TDVP formulations, even those that eliminate small eigenvalues of the Fisher matrix or use geometric properties to recast the equation of motion. We conclude that alternative methods need to be developed to leverage the computational efficiency of explicit time integration of the TDVP equations for simulating strongly nonequilibrium quantum dynamics with neural-network quantum states.

# 1   Introduction

Access to quantum dynamics of many-body systems is of key relevance to many research fields in physics and chemistry, and in particular, in condensed matter [1]. For problems that usually appear in these fields, accessing large quantum systems is key to a more complete understanding of quantum phenomena. However, access to large quantum systems is rendered notoriously difficult by the exponential growth of the Hilbert space [2, 3]. Recently, Neural Quantum States (NQS) have emerged as powerful methods that bypass the limitations imposed by existing methods [4–6], showing fascinating results for some of the most challenging 2D systems [7–10]. However, the results reported so far are limited to dynamics near the ground state, and strongly driven quantum dynamics are generally considered highly challenging [11–15].

A standard approach to obtaining dynamics from a variational representation is the Time-Dependent Variational Principle (TDVP) [16–19], first formulated by Dirac. TDVP generates a system of nonlinear ordinary differential equations (ODE) of motion for the variational parameters, which can be solved with standard explicit integration schemes [20], providing access to large systems [7–10]. However, this ODE system requires Monte Carlo sampling of the energy and wave function gradients [21] entering the equations, making it prone to stochastic noise. Typically, the combination of noise and nonlinearities leads to a buildup of numerical errors [11, 22, 23], eventually destabilizing the integration. Stochastic sampling of states with a small contribution to the wave function can also lead to wrong estimations of parameter updates [24]. In addition, the variational representation of the quantum wave function is singular in almost all cases, which makes TDVP equations mathematically ill-defined [25, 26]. A standard trick for this issue is to introduce a regulator: a mathematical artefact that can also lead to numerical instabilities [7, 10, 11, 13, 27]. However, it remains unclear if the above-mentioned sources of instabilities are necessary and sufficient to explain the reported numerical breakdowns and truly limit simulating strongly driven quantum dynamics.

Recently, several works have moved towards improving the NQS time evolution by exploring alternatives to the standard TDVP. One approach, introduced in [28] and [24], directly optimizes the overlap between variational and exact evolution at each time with Suzuki-Trotter decomposition, which is then used to calculate the forward-propagation gradient of variational parameters. In [23, 29], forward-propagation and integration are replaced with learning the entire quantum time evolution globally. In [14], the time evolution is obtained with the implicit midpoint integrator, while optimizing the error between the variational and exact propagation at each time step. Yet another approach utilizes the autoregressive property of neural networks to obtain stable time evolution through normalized wave function representations [22]. However, compared to the standard realization of TDVP, these approaches require significantly more computational efforts, while the actual dynamics might not even require such complexity. Furthermore, the use of these advanced methods still does not guarantee that all sources

of numerical instabilities are addressed, even for simple dynamical scenarios.

Therefore, in this paper, we approach the problem of numerical instabilities differently. We seek to identify the origin of the problematic part of the TDVP time integration, rather than replace it with a more complex method. Attempts at this could already be drawn from the literature. For example, [30] introduced a taming scheme that can be used to rescale the energy gradient in TDVP, which could stabilize the integration if the gradient becomes too large. Additionally, several reformulations of TDVP have been introduced, like working in the regular subspace of the diagonalized Fisher matrix [8, 31], or using the geometry of the variational manifold to recast TDVP equations [25]. However, a systematic comparison between these improvements to the standard method is missing. Similarly, assessing the importance of the mentioned sources of instabilities and potentially identifying new ones is missing as well. Thus, we critically assess the possible sources of inaccuracy that can appear in TDVP using a numerically cheap explicit time integration method. As a benchmark, we compare these results with exact diagonalization (ED) and implicit time integration. In all cases, we remove the sampling noise from the TDVP equation of motion by calculating the quantum averages over the whole Hilbert space. Furthermore, we study the significance of regularization by comparing it with two additional formulations of TDVP designed to make the integration regular. Finally, to assess the impact of singularity on time integration, we observe the spectrum of the quantum geometric tensor.

This paper is organized as follows. In Sec. 2, we describe the Hamiltonian and the neural network models, as well as all the TDVP formulations and integrators. In Sec 3, we present the correlation dynamics obtained by integrating the TDVP equation of motion for various driving amplitudes and with various methods, unveiling an undocumented numerical breakdown regime. In Sec. 4, we discuss the results and their implications. Finally, we conclude and provide an outlook in Sec. 5.

## 2 Methods

In this work, we study a system of antiferromagnetically coupled quantum spins on a 2D lattice, interacting according to the nearest-neighbour Heisenberg model:

$$H = J_0 \sum_{\{i,j\}\in X,Y} \mathbf{S}_i \cdot \mathbf{S}_j + J(t) \sum_{\{i,j\}\in Y} \mathbf{S}_i \cdot \mathbf{S}_j, \tag{1}$$

where $\mathbf{S}_i$ is the spin operator on the $i$-th site. The sum over $\{i, j\} \in X, Y$ is taken over all nearest-neighbour pairs of the lattice, while $\{i, j\} \in Y$ indicates perturbation of vertical bonds by the function $J(t)$. This is a minimal model to represent the terahertz dynamics of magnetic systems driven by an optical perturbation of exchange interactions [32–35]. The system is prepared in the ground state and driven by a quench-like perturbation:

$$J(t) = \begin{cases} 0, & t < 0 \\ \Delta J_0, & t \geq 0 \end{cases}. \tag{2}$$

As a variational representation, we use the archetypical Restricted Boltzmann machine [7]:

$$\Psi(s) = \prod_{j=1}^{M} 2\cosh(\theta_j(s)). \tag{3}$$

Here, $s = \{s_i^z\}$, $i = 1, \ldots, N$ is the spin configuration of $N$ particles, $M = \alpha N$ determines the expressive power of the network parametrized by $\alpha$, and $\theta_j = b_j + \sum_i s_i^z w_{ij}$ includes the

biases $b_j$ and weights $w_{ij}$ of the network. This network has been successful in representing a wide variety of quantum spin models [6, 7], leveraging physical symmetries to reduce the number of network parameters [7, 36–38], providing access to large systems. Examples include the ultrafast dynamics in the antiferromagnetic Heisenberg model [9, 11, 26, 38, 39], and the transverse-field Ising model [7, 8, 10]. The time dependence of the neural network is encoded in the time dependence of its parameters. These follow the TDVP equation of motion [16, 17]:

$$S_{kk'}\dot{W}_{k'} = -iF_k, \tag{4}$$

where elements $W_{k'}$ include all the RBM parameters. The elements $F_k = \langle E_{\text{loc}}O_k^* \rangle - \langle E_{\text{loc}} \rangle \langle O_k^* \rangle$ constitute the energy gradient vector in the parameter space. The covariance matrix elements $S_{kk'} = \langle O_k^* O_{k'} \rangle - \langle O_k^* \rangle \langle O_{k'} \rangle$ define the quantum Fisher matrix (QFM) [26], which is the metric of the parameter space of the selected network [25, 27]. Here, $\langle \cdot \rangle$ represents the quantum-mechanical average over the entire Hilbert space. The logarithmic derivative functions are defined as $O_k(s) = 1/\Psi(s) \cdot \partial_{W_k}\Psi(s)$, while the local value of energy is $E_{\text{loc}}(s) = \langle s | \hat{H} | \Psi \rangle / \Psi(s)$.

The TDVP equation consists of a set of first-order differential equations for network parameters. Since these equations are nonlinear, even for the simplest neural network architectures, numerical integration is unavoidable to solve them. For this task, we consider three different formulations of TDVP, which we refer to as *regularization* [3, 21], *diagonalization* [8, 31], and the *geometric method* [25]. We use these formulations to solve the TDVP equation of motion with an explicit integrator, and also compare this with implicit integration. To describe what these methods do, we rewrite the TDVP equation of motion as $\dot{\mathbf{W}} = \mathbf{f}(\mathbf{W})$, where a vector is defined in bold as $\mathbf{W} = (W_1, W_2, \dots)$. Here, $\mathbf{W} = \mathbf{W}(t)$ is the vector of all network parameters, and the update function $\mathbf{f}$ is obtained by solving Eq. (4) at some time $t$. Then, an integrator defines a way to calculate the parameter vector in the next integration step $\mathbf{W}_{p+1} = \mathbf{W}(t + dt)$, using the current $\mathbf{W}_p = \mathbf{W}(t)$. A formulation defines a way to obtain the update $\mathbf{f}$. Formally, the update function is the inverse of Eq. (4):

$$f_k = -iS_{kk'}^{-1}F_{k'}. \tag{5}$$

The $S$-matrix is singular in general, and therefore non-invertible, which means that Eq. (5) denotes a Penrose-Moore pseudoinverse [40].

An overview of integrators and formulations used in this work is given in Table 1. The explicit integrator, and in particular the Heun's scheme [26, 27], is a standard in TDVP time integration, where $\mathbf{W}_{p+1}$ can be directly calculated from $\mathbf{W}_p$. In contrast, the implicit midpoint update [14] cannot be solved directly, so a root-finding algorithm must be used. We used the Newton-Raphson method [41] to solve Eq (7), implemented in the SciPy package [42]. Compared to explicit schemes, implicit integration is more accurate, but also more computationally expensive. The three formulations we used each provide a different way to deal with the singularity of the $S$-matrix in Eq. (5). Regularization is considered a standard, and it offsets the matrix diagonal by a small value; diagonalization solves the TDVP equation (4) in the diagonal basis of $S$; the geometric method uses the geometric properties of the variational manifold to recast Eq. (4) into a linear problem. More details on these formulations can be found in the Appendix B. Note that one can use any combination of integrator and formulation to solve the TDVP equation of motion. Finally, we introduce another modification to the TDVP equation of motion: taming, explained in Appendix C. This procedure rescales the gradient of the update function in Eq. (5) to control the influence of the nonlinearity of equations of motion in creating numerical instabilities.

Our aim is to first evaluate the accuracy of TDVP integration in the standard scheme, with regularization and explicit integrator, for different quench strengths $\Delta$ in Eq. (2). Next, we explore the possibility of regulator-free time integration by using diagonalization and the geometric method with the explicit integrator. We further assess how explicit integration compares

Table 1: Integrators and formulations used to solve the TDVP equation of motion (4). A more detailed overview of formulations can be found in Appendix B. In regularization, $\mathbb{1}$ is a unit matrix. For diagonalization, $\text{Diag}(S)$ denotes the $S$-matrix in its diagonal basis, and subscripts zero and nonzero denote the singular and the nonsingular subspaces of the basis, respectively. In the geometric method, $\mathbf{f}_{\text{geo}}$ and $\lambda$ form a vector which solves Eq. (10), and the Lagrange multipliers $\lambda$ are discarded.

| Integrators | |
|---|---|
| Explicit Heun's scheme | $\mathbf{W}_{p+1} = \mathbf{W}_p + \dfrac{\mathrm{d}t}{2}\Big(\mathbf{f}(\mathbf{W}_p) + \mathbf{f}\big(\mathbf{W}_p + \mathrm{d}t\mathbf{f}(\mathbf{W}_p)\big)\Big)$    (6) |
| Implicit midpoint | $\mathbf{W}_{p+1} = \mathbf{W}_p + \mathrm{d}t\mathbf{f}\Big(\dfrac{1}{2}\big(\mathbf{W}_p + \mathbf{W}_{p+1}\big)\Big)$    (7) |

| Formulations | |
|---|---|
| Regularization | $S \to S_{\text{reg}} = S + \varepsilon\mathbb{1}$ <br> $\mathbf{f}_{\text{reg}} = -iS_{\text{reg}}^{-1}\mathbf{F}$    (8) |
| Diagonalization | $S \to \text{Diag}(S) = S_{\text{zero}} \oplus S_{\text{nonzero}}$ <br> $\mathbf{F} \to \mathbf{F}_{\text{zero}} \oplus \mathbf{F}_{\text{nonzero}}$ <br> $\mathbf{f}_{\text{dia}} = -iS_{\text{nonzero}}^{-1}\mathbf{F}_{\text{nonzero}}$    (9) |
| Geometric method | $S \otimes \begin{pmatrix} 1 & i \\ -i & 1 \end{pmatrix} = S_{\text{geo}} \begin{cases} g = \text{Re}S_{\text{geo}} \\ \omega = \text{Im}S_{\text{geo}} \end{cases}$ , $\mathbf{F} \otimes \begin{pmatrix} 1 \\ -i \end{pmatrix} = \mathbf{F}_{\text{geo}}$ <br><br> $\begin{pmatrix} 2g & \omega^T \\ \omega & 0 \end{pmatrix}\begin{pmatrix} \mathbf{f}_{\text{geo}} \\ \lambda \end{pmatrix} = \begin{pmatrix} 0 \\ -\mathbf{F}_{\text{geo}} \end{pmatrix}$    (10) |

to implicit integration in all formulations, especially for the task of handling numerical instabilities. We track the accuracy and the stability of the time-integration methods by observing: (i) the nearest-neighbour correlation function on a quenched bond, and (ii) the spectrum of the S-matrix. The former captures the leading-order dynamics of observables [35]; thus, we use it to measure accuracy. The latter tells how problematic the $S$-matrix singularity is in performing time integration [31].

Most of the analysis is carried out on a small $2 \times 2$ lattice. A simple system like this one is small enough to have access to exact diagonalization, which we use as a benchmark for NQS solutions. Contrary to the exact wave function, we obtain the variational wave function by integrating the TDVP equation of motion. Here, we again take advantage of the small system size to calculate quantum averages over the full Hilbert space, which rules out errors due to sampling noise. In addition, we also calculate the dynamics of bigger lattices and network architectures sampled with variational Monte Carlo (VMC) [21], using the ULTRAFAST numerical package [43]. The time integration in ULTRAFAST is done with the standard approach:

explicit Heun's scheme (6) as integrator, and regularization (8) as formulation. We compare the results of a small system summed over the full Hilbert space to those of bigger, Monte Carlo sampled systems in the same physical setting. This allows us to assess if the same observations apply to bigger systems, wider networks, and stochastic sampling.

In all calculations, the system is initialized in the ground state of the Heisenberg model (1). For NQS variational representations, the ground state is found by a gradient descent algorithm starting from random network parameters. The sign structure of the ground-state wave function is known to obey Marshall's sign rule [44], which can be enforced by a gauge transformation of the Hamiltonian [26]. The sign rule is obeyed with real network parameters; therefore, we initialize them as real.

## 3  Results

In this section, we present the numerical time integration results of an antiferromagnetic $2 \times 2$ lattice represented by the RBM neural network. Since our goal is to explore the possibility of computationally efficient time integration of TDVP, we use a simple RBM architecture with only one hidden node. This corresponds to a $\alpha = 1/4$ architecture, which has the same number of parameters as $\alpha = 1$ when translational invariance is taken into account. We compare the results of numerical time integration with exact results.

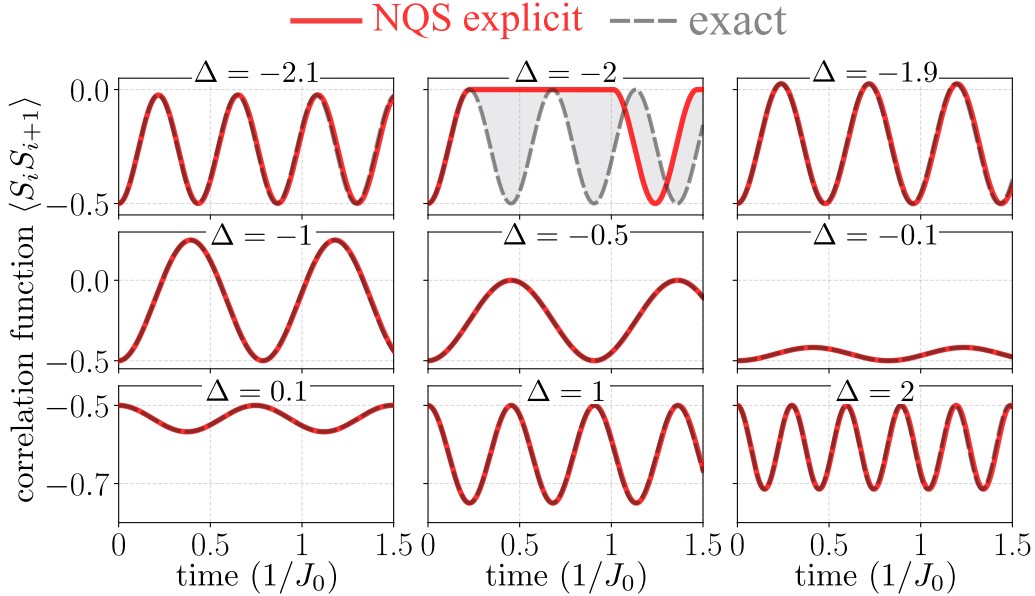

Figure 1: Correlation dynamics of NQS compared to ED, as a function of quenching strength $\Delta$. Red lines indicate NQS results, while the dashed gray lines are obtained with ED. Quench strengths are shown on top of each graph. In almost all cases, NQS results agree very well with ED, except for the breakdown quench of $\Delta = -2$.

We first showcase the dynamics of the correlation function $C_{ij} = \langle S_i \cdot S_j \rangle$ of two spin sites $i$ and $j$ on a quenched vertical bond, as a function of quench parameter $\Delta$ in Eq. (2). To obtain the dynamics, we used a standard approach: the TDVP equation of motion was solved at each time $t$ by regularization formulation Eq. (8), and integrated with Heun's update rule Eq. (6). Results of this analysis can be found in Fig. 1. Given the success of the NQS method in various physical scenarios, it is not surprising that the NQS time integration shows an excellent agreement with ED results for almost all values of $\Delta$. This is consistent across various frequencies and amplitudes of correlation oscillations. However, we identify a specific value of $\Delta = -2$,

where a numerical breakdown happens. Specifically, when the correlation function reaches the first maximum, all dynamics stop. We refer to this *breakdown point* as a problematic scenario where the explicit TDVP time integration is unable to recover the correct correlation dynamics.

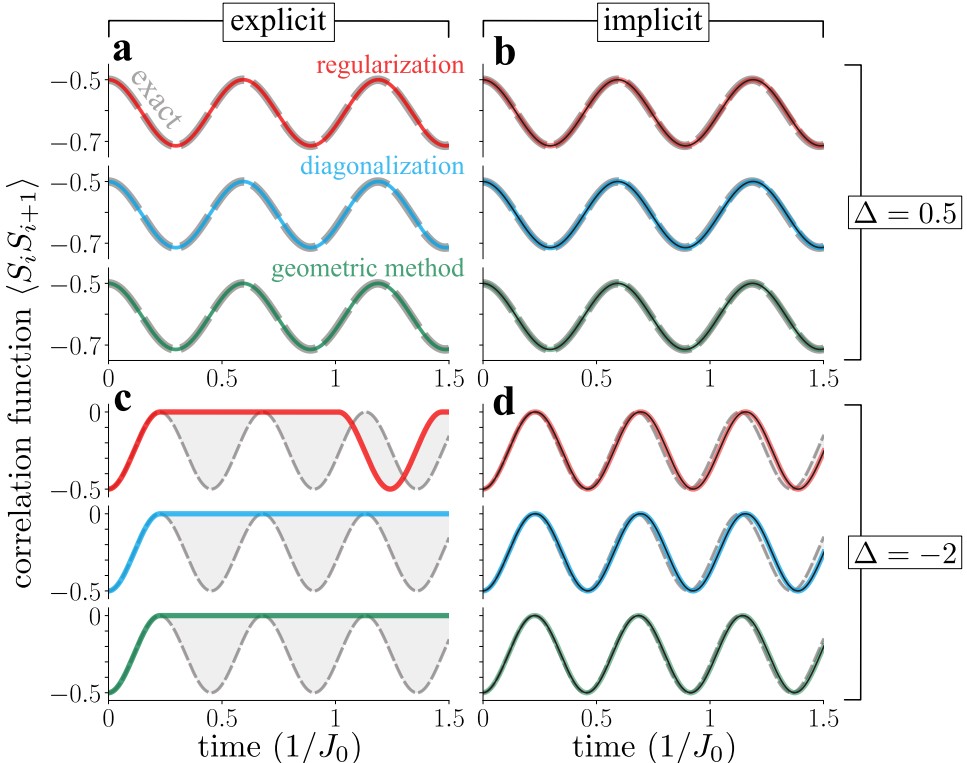

Figure 2: Dynamics of correlation function between neighbouring spins along a quenched bond as a function of time, for two quenches: (a, b) top row $\Delta = 0.5$, and (c, d) bottom row $\Delta = -2$. All full lines are obtained by TDVP time integration of NQS represented by the RBM architecture in Eq. (3) with $\alpha = 1/4$. Dashed gray lines are the ED results. Colors correspond to different formulations: *regularization* (red), *diagonalization* (blue), *geometric method* (green). All NQS results for the well-behaved $\Delta = 0.5$ agree with the ED results. For the breakdown quench $\Delta = -2$, explicit integrator (left) produces wrong results in all formulations, while the implicit integrator (right) produces correct dynamics. The regularization curve in (c) contains a region of interruption from the frozen dynamics, but still does not recover the correct result.

As an alternative to the standard approach, we now present results obtained by other formulations of the TDVP equation of motion and the implicit integrator. The results are shown in Fig. 2 for two quenches: $\Delta = 0.5$ and $\Delta = -2$, and for the combination of both integrators and all formulations from Table 1. We chose a well-behaved quench strength $\Delta = 0.5$ to demonstrate that accurate integration is possible, even without regularization, and using both integrators. All NQS results in the top row of Fig. 2 follow the exact results. However, in the bottom row, the breakdown quench $\Delta = -2$ shows two qualitatively different behaviours. For explicit integration, the breakdown persists regardless of the formulation used. Therefore, a choice of formulation plays no role in correctly calculating the dynamics for this quench for explicit integration. When we change the integrator to implicit, the accuracy of integration is greatly increased, and correct dynamics are recovered. It should be noted, though, that for $\Delta = -2$, the combination of geometric formulation and implicit integration is the most

accurate, as visible by the smallest offset from the exact curve. Thus, for further considerations about the implicit integrator, we used this formulation.

Next, we present the extension of the breakdown analysis to wider networks and larger lattices. We follow the same recipe as for the $2 \times 2$ lattice with $\alpha = 1/4$ RBM network architecture. Starting from the ground state of the model, we quench the vertical bonds of the lattice with a $\Delta = -2$ strength, and integrate the TDVP equation of motion with Heun's scheme, in the regularization formulation. These results were obtained with the ULTRAFAST package, where variational Monte Carlo is used to sample the quantum expectation values, unlike the approach shown so far. In Fig. 3, we show the dynamics of the correlation function for $\alpha = \{1, 2, 3, 4, 5\}$ for the small $2 \times 2$ lattice, as well as $4 \times 4$ and $6 \times 6$ lattices with $\alpha = 1$. In all cases, the number of independent parameters is reduced by exploiting the translational symmetry of the lattice. As indicated by the results, the breakdown regime persists across different network architectures and different lattice dimensions, for the same perturbation strength. We also tested bigger network widths $\alpha$ for $4 \times 4$ and $6 \times 6$ lattices, but the results show the same behaviour.

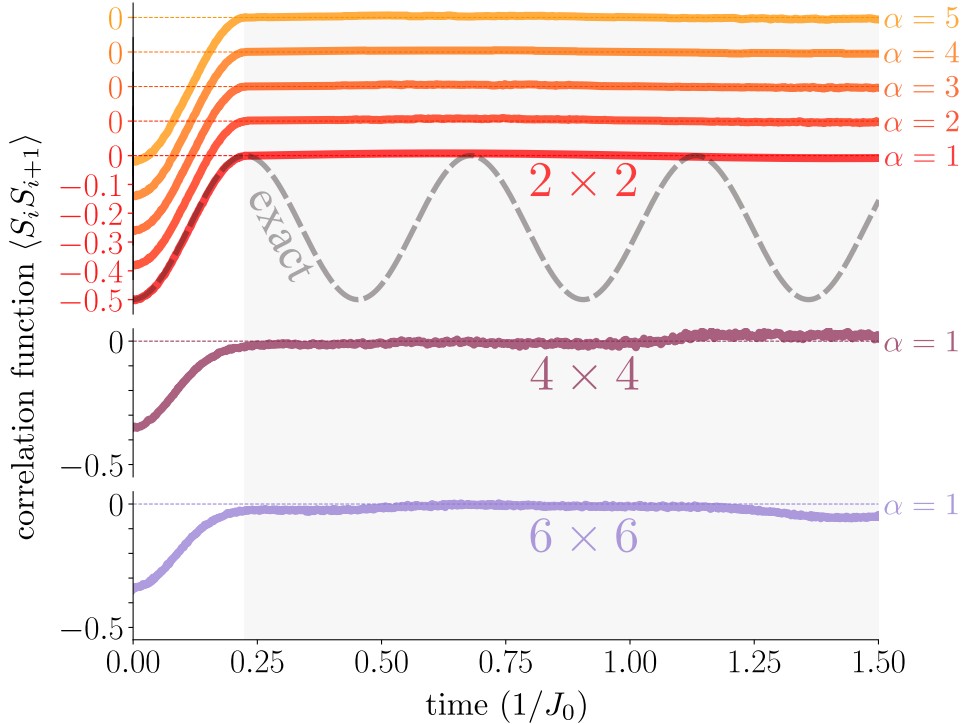

Figure 3: Correlation dynamics similar to results in Fig. 2, for the same value $\Delta = -2$ of quench strength, but for bigger systems and wider neural networks. On the top graph, results are shown for the $2 \times 2$ lattice, and different values of the network width parameter: $\alpha \in \{1, 2, 3, 4, 5\}$. The dashed grey line refers to the correct result obtained by ED. The two bottom graphs show the same for $4 \times 4$ and $6 \times 6$ lattices, with $\alpha = 1$. All simulations show a numerical breakdown for this quench, characterized by a loss of dynamics after the first maximum of the correlation function (indicated by the shaded area).

To study the effect of singularity on time integration, we observe the spectrum of the $S$-matrix. This matrix is interpreted as a metric tensor of the parameter space [25, 27], so the TDVP equation guides the parameters along a geometrically optimal trajectory. However, if the matrix has zero eigenvalues, there are directions where the evolution of trajectories is unconstrained by the metric. This can lead to numerical instabilities, especially if the trajectory

obtains components in these redundant directions. Specifically, we are interested in whether there are new emerging singular directions at the breakdown point. Thus, to assess how the singularity influences the stability of time evolution, we study the spectrum of the $S$-matrix as a function of time. The results are presented in Fig. 4 for: (i) explicit integration by Heun's scheme, (ii) implicit midpoint integration, (iii) exact solutions obtained by infidelity optimization. More details calculating the exact RBM representation using infidelity [22, 31, 45] can be found in Appendix A. First, we indicate that there are always eigenvalues with values at zero in machine precision, regardless of the integration method, marked as "vanishing eigenvalues". These originate from the overparametrization of the NQS representation. There are also eigenvalues denoted as "finite", whose values are never small throughout the dynamics, so they pose no problem for integration. Secondly, and more interestingly, some eigenvalues occasionally have small values for all the presented methods. These small eigenvalues range from $10^{-9}$ to $10^{-3}$ orders of magnitude, still significantly larger than the machine precision. In particular, the implicit method shows cusps at times coinciding with correlation maxima in Fig. 2 (c. d), the first of which is the breakdown point. The depth of these cusps, or the smallest nonzero eigenvalue, is shown on the inset as a function of integration time step. Decreasing the time step makes the implicit method's cusps reduce to smaller values, saturating around $10^{-9}$. Explicit integration and exact fits are largely unaffected by the reduction of the time step. It should be noted that the dynamics of the correlation function in Fig. 2 are unaffected by the reduction of the time step for all methods presented in this paper.

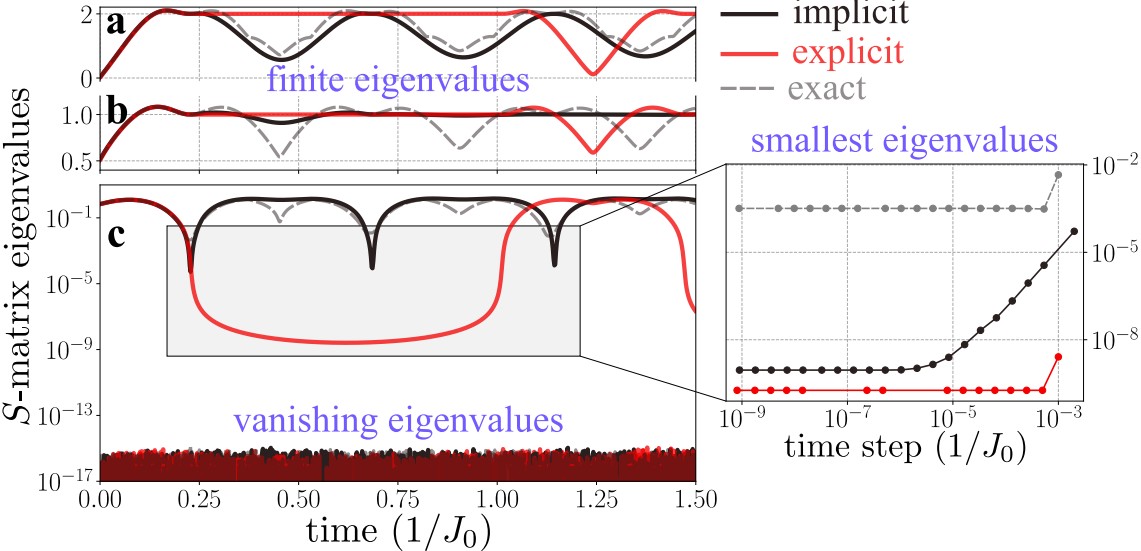

Figure 4: Spectra of the Quantum Fisher Matrix Eq. (4). The eigenvalues were obtained using three different methods: implicit integration of TDVP using Eq. (7) and the geometric method formulation Eq. (10), explicit integration of the TDVP using Eq. (6) and the regularization formulation Eq. (8), and fitting the RBM architecture to ED solutions using infidelity optimization. The latter is referred to as exact. Three distinct classes of eigenvalues are observed. Finite eigenvalues (a, b) always have a well-behaved value that doesn't cause a singularity. Vanishing eigenvalues (c) always have a value close to zero in machine precision. On the bottom graph, we also see eigenvalues whose value occasionally becomes small for all methods shown. The inset shows the smallest eigenvalues as a function of integration time step, indicating the dependence of eigenvalue cusp depth as a function of numerical accuracy.

# 4   Discussion

The numerical breakdown we introduced in the previous section occurs only at the perturbation parameter $\Delta = -2$. In our analyses, we have not observed an error onset in any quantum observable that would indicate an emergence of the breakdown regime as a function of perturbation. This is visible from the results in the top row of Fig. 1, where two quenches very close in value to the breakdown quench still produce correct dynamics. We stress that all well-behaved results have been obtained by explicit integration in a standard formulation. Furthermore, the NQS numerical error is systematically reducible by reducing the time step for all quenches except at the breakdown. Thus, the origin of the breakdown remains elusive. Even though a quench of this magnitude is physically unrealistic, the cause of the breakdown is not physical, as indicated by the well-behaved ED results. Interestingly, the breakdown is not caused by either stochastic noise or the artificial nature of the regulator. The former is demonstrated by summing over the entire Hilbert space, thus completely removing the sampling noise. The latter is deduced from using two formulations, *diagonalization* and *geometric method*, both of which effectively remove the need for regularization, but still retain the breakdown.

We emphasize that, even in this regulator-free approach to numerical time integration, with full summation over the Hilbert space, a previously unobserved breakdown regime emerges. This challenges the current understanding of numerical breakdown in variational representations, where instabilities are believed to originate from the interplay between noise, nonlinearity, and singularity. Note that the breakdown could still be caused by the nonlinearity of equations of motion. We introduced taming (Appendinx C) to deal with this issue, but that might not be sufficient. We also noticed no drastically varying time scales of dynamics, indicating no stiffness, which could otherwise cause numerical instabilities.

As shown in Fig. 2, the only way we could treat the breakdown was by using the implicit midpoint integrator. Alongside Heun's scheme, we also attempted the Runge-Kutta fourth-order (RK4) scheme, which gave the same results. We also performed an extensive time step analysis for both Heun's and RK4 schemes, which again produced the breakdown for smaller time steps. This demonstrates the failure of explicit schemes to capture the correct dynamics in the breakdown scenario, and indicates that explicit schemes with adaptive time steps [8] still do not recover the correct solution. On the other hand, implicit schemes work by minimizing the overlap between the left- and right-hand sides of the equation of motion, thus greatly increasing the stability of time integration. However, this minimization comes at great computational costs, as the elements of the TDVP equation of motion have to be evaluated at every trial step. This makes the implicit midpoint method very inconvenient to scale to larger systems.

As far as the scaling properties are concerned, we have shown results for wider networks and bigger lattices in Fig. 3. We consistently see the presence of the breakdown regime across all the scaling parameters. Network width is controlled by the parameter $\alpha$ in Eq. (3), which determines the network's expressive power by setting the number of its hidden neurons. We thus conclude that the breakdown was not caused by an insufficiently expressive network. In fact, the results from implicit integration show that it's possible to represent the correct results, even with low expressivity, with only one RDM hidden node. To further support this claim, we indicate that we were able to fit the network to the correct ED dynamics, as explained in Appendix A, for the same expressive power. In addition, we observe that the same quenching parameter consistently produces the breakdown across various lattice sizes. Breakdown due to the $\Delta = -2$ quench is therefore not a finite-size effect, but rather a universal phenomenon for bigger lattices as well.

Finally, we address the singularity of the $S$-matrix in Fig. 4. For all methods, finite eigenvalues don't introduce any difficulties in time integration, and the singularity universally present

due to overparametrization can be circumvented. We thus focus on the smallest nonzero eigenvalues, shown in Fig. 4 (c). Their values for the exact representation are around $10^{-3}$ order of magnitude, and $10^{-9}$ for the explicit integration. Interestingly, around the breakdown point, the implicit integration yields eigenvalues within the same range, depending on the time step of integration. However, regardless of the time step and therefore the order of magnitude of the smallest nonzero eigenvalue, implicit integration always recovers the dynamics correctly. In contrast, the explicit integration, whose minimal eigenvalue is very close to the saturated value of the implicit case, never recovers the proper dynamics. So, having two cases of very small eigenvalues for two different integrators with distinct accuracies, we conclude that the magnitude of the smallest eigenvalue does not determine the success of time integration. The eigenvalues are still orders of magnitude larger than machine precision, and this is not problematic for numerical calculations in the absence of Monte Carlo noise. Therefore, there is no additional singularity that emerges with time, which would make the explicit integration problematic. Furthermore, the behaviour of small eigenvalues is qualitatively similar for implicit integration and the exact representation. In both methods, eigenvalues have cusps around the times of correlation maxima, while the exact representation has cusps even around the correlation minima, unlike the implicit method. Since these methods both recover the correct dynamics, the magnitudes of the smallest eigenvalues don't seem to be an obstructive factor in the success of time integration. Collecting these observations, we conclude that the time integration is possible regardless of the singularity of the $S$-matrix, and the breakdown does not emerge from the singularity.

## 5   Conclusion

To conclude, we have presented an analysis of the stability and accuracy of NQS, which represents quenched dynamics of the Heisenberg antiferromagnet with TDVP. We uncovered a numerical breakdown, even with a fully sampled Hilbert space, and without regularization. Interestingly, the breakdown does not originate from known problematic factors in TDVP, showing that numerical time integration with explicit methods is more challenging than anticipated before. Still, our conclusions are limited to the Restricted Boltzmann Machine architecture. For the presented physical scenario, other network architectures could prove to be better variational ansätze, such as a Deep Boltzmann Machine [46], or the transformer [23,47,48]. Both of these networks have already had great success in the NQS field. In addition, a deeper understanding of the breakdown origin is required to find a way to treat it. With this paper, we hope to motivate a search for a different approach in obtaining stable and accurate NQS dynamics, free of breakdowns, computationally cheap, and scalable to bigger systems. A path towards this may come from restricting the integration to a more stable part of the variational manifold, such as the normalized subspace [22], or reformulating TDVP beyond the current order of expansion [21]. Further research may focus on constructing a hybrid explicit-implicit scheme to handle unstable points with more expensive integration techniques. In addition, more research is needed to elucidate the role of $S$-matrix eigenvalue cusps in the failure of explicit integration.

## Acknowledgements

This project was funded by the NIMFEIA: Nonlinear Magnons for Reservoir Computing in Reciprocal Space project of the European Union, under the number 101070290. JHM acknowledges funding by the Dutch Research Council (NWO) via VIDI project number 223.157

(CHASEMAG). We are grateful to Mikhail Tretyakov and Gabriel Lord for helpful discussions regarding the stability of numerical time integration methods.

# A  Infidelity optimization of a variational wave function

To obtain the variational representation $|\phi\rangle = |\Psi_{\mathrm{RBM}}(\mathbf{W})\rangle$ of the exact diagonalization wave function $|\psi\rangle = |\Psi_{\mathrm{ED}}(t)\rangle$, we optimize the RBM architecture using infidelity as loss function:

$$L(\psi, \phi) = 1 - \frac{\langle\psi|\phi\rangle\,\langle\phi|\psi\rangle}{\langle\psi|\psi\rangle\,\langle\phi|\phi\rangle}, \tag{A.1}$$

which is a measure of overlap between two vectors. Here, we fix the exact wave function at some time $t$, while varying the parameters $\mathbf{W}$ of the variational ansatz. The update rule for variational parameters is the gradient descent variant:

$$\mathbf{W}(p+1) = \mathbf{W}(p) - \eta S^{-1}\nabla_{\mathbf{W}}L, \tag{A.2}$$

where $p$ is the optimizaiton step, and $\eta$ the learning rate. Here, $S^{-1}$ is the inverse of the Quantum Fisher matrix, defined in the same way as in Eq. (4), and inverted using the regularization procedure described in B.1. We calculate the infidelity gradient $\nabla_{\mathbf{W}}L$ using:

$$\nabla_{W_k}L = \frac{\langle\Psi_{\mathrm{ED}}|\Psi_{\mathrm{RBM}}\rangle \cdot \left(\langle O_k^*\rangle\,\langle\Psi_{\mathrm{RBM}}|\Psi_{\mathrm{ED}}\rangle - \sum_{s_i}O_k^*(s_i)\Psi_{\mathrm{RBM}}^*(s_i)\Psi_{\mathrm{ED}}(s_i)\right)}{\langle\Psi_{\mathrm{RBM}}|\Psi_{\mathrm{RBM}}\rangle}. \tag{A.3}$$

We've also taken into consideration that $|\Psi_{\mathrm{ED}}\rangle$ is normalized. In Eq. (A.3), the logarithmic derivative $O_k(s)$ is the same as described in Eq. (4), and $s_i$ is the $i$-th spin configuration in the Hilbert space.

# B  TDVP formulations

In this appendix, we describe how to solve the TDVP equation of motion using three different formulations.

## B.1  Regularization

In the most general situation, the $S$-matrix in Eq. (4) is singular, which prevents the inversion of the equation. The standard approach to deal with this obstacle is regularization: $S \to S_{\mathrm{reg}} = S + \varepsilon\mathbb{1}$, where $\mathbb{1}$ is the unit matrix, and $\varepsilon$ is a small number, often in the interval $\left[10^{-5}, 10^{-3}\right]$. Following the regularization approach, the recipe for finding the elements of the update function $\mathbf{f}(\mathbf{W})$ in Eq. (5) is:

$$\mathbf{f}_{\mathrm{reg}} = -i S_{\mathrm{reg}}^{-1}\mathbf{F}. \tag{B.1}$$

Introducing this small diagonal offset makes the determinant $\det(S_{\mathrm{reg}})$ finite, thus rendering the inverse $S_{\mathrm{reg}}^{-1}$ well-defined. However, Hofmann et. al. [11] demonstrated a fine balance between stability, accuracy, and regularization, which is especially delicate if the numerical method has a stochastic component, like the Monte Carlo method. Improper choice of regularization can lead to numerical instabilities and eventual breakdowns, occurring sooner in dynamics for stronger perturbations. Therefore, due to known problems with regularization, here we formulate regulator-free integration using two alternative approaches for finding the update function in Eq. (5).

## B.2 Diagonalization

This approach follows from the fact that singular matrices have zero eigenvalues, which don't contribute to dynamics. Thus, the first step of the diagonalization method is diagonalizing the $S$-matrix and obtaining the eigenspace transformation matrix $T$. The elements of the TDVP equation of motion are then transformed: $S \rightarrow S_{\mathrm{dia}} = \mathrm{Diag}(S) = T^{-1}ST$, $\mathbf{F} \rightarrow \mathbf{F}_{\mathrm{dia}} = T^{-1}\mathbf{F}$, $\mathbf{W} \rightarrow \mathbf{W}_{\mathrm{dia}} = T^{-1}\mathbf{W}$. The final step is to remove the nullspace obtained from the transformation matrix $T$ by removing elements that correspond to zero eigenvalues, from all terms of the equation. This removal is done numerically, following a predetermined criterion that eigenvalues below a certain value $\zeta$ are considered to be zero. The deletion produces elements $S_{\mathrm{nonzero}}$ and $\mathbf{F}_{\mathrm{nonzero}}$ in Eq. (9).

The removal of nullspace boils down to just ignoring those elements of $S_{\mathrm{dia}}$, $\mathbf{F}_{\mathrm{dia}}$, and $\mathbf{W}_{\mathrm{dia}}$ that have the indices of zero eigenvalues. This effectively reduces the dimension of the inversion problem. We find that the zero cutoff criterion $\zeta \approx 10^{-12}$ usually works well to leave the $S_{\mathrm{nonzero}}$ matrix regular. This way, the equation can be inverted and the update function can be calculated without regularization:

$$\mathbf{f}_{\mathrm{dia}} = -iS_{\mathrm{nonzero}}^{-1}\mathbf{F}_{\mathrm{nonzero}}, \tag{B.2}$$

after which, the problem is transformed back into the original parameter basis. Note that if a numerical integrator uses multiple steps, like the Heun update rule Eq. (6), the diagonalization procedure described here is required at each intermediate step.

## B.3 Geometric method

This approach is based on the variational methods formulation on a Kähler manifold, introduced by Hackl et. al. in [25]. The core of the method is to reparametrize the TDVP equation of motion into a problem where all the parameters of the neural network are real by splitting them into real and imaginary components. We thus perform the following transformation:

$$\mathbf{W} \rightarrow \mathbf{W}_{\mathrm{geo}} = \{\mathrm{Re}W_1, \mathrm{Im}W_1, \ldots, \mathrm{Re}W_M, \mathrm{Im}W_M\}, \tag{B.3}$$

which doubles the dimension of the problem. Given this transformation and the properties of the RBM network Eq. (3), the transformation rule for other elements of the TDVP equation of motion is:

$$S, \mathbf{F} \rightarrow S_{\mathrm{geo}} = S \otimes \begin{pmatrix} 1 & i \\ -i & 1 \end{pmatrix}, \ \mathbf{F}_{\mathrm{geo}} = \mathbf{F} \otimes \begin{pmatrix} 1 \\ -i \end{pmatrix}. \tag{B.4}$$

After transforming all the elements of the equation, two geometric characteristics of the variational manifold are defined: the metric $g = \mathrm{Re}S_{\mathrm{geo}}$, and the symplectic form $\omega = \mathrm{Im}S_{\mathrm{geo}}$. These are used to define the new TDVP equation of motion. In addition, following the prescription in [25], we can recast the pseudoinversion problem required to solve Eq. (4) into:

$$\underbrace{\begin{pmatrix} 2g & \omega^T \\ \omega & 0 \end{pmatrix}}_{A}\underbrace{\begin{pmatrix} \mathbf{f}_{\mathrm{geo}} \\ \lambda \end{pmatrix}}_{x} = \underbrace{\begin{pmatrix} 0 \\ -\mathbf{F}_{\mathrm{geo}} \end{pmatrix}}_{B}, \tag{B.5}$$

where $\lambda$ is a vector of Lagrange multipliers. The multipliers serve to constrain the TDVP equation of motion to a subclass of solutions that have a minimal-length component in the singular subspace of the matrix $\omega$. This constraint is meant to eliminate potential instabilities caused by the singularity of the matrix.

This is now a linear problem in the form $Ax = B$ whose dimension is four times the dimension of the original formulation. However, even though the network redundancies that cause

the singularity of the $S$-matrix are now recast, the problem still contains them in the matrix $A$. Therefore, to find a solution to this linear problem, we must use a convergence method which minimizes the distance between the left- and right-hand sides of the equation (B.5). For this task, we used the least-squares algorithm [49] implemented in NumPy [50], which finds the $x$ that minimizes the norm $\|Ax - B\|$ in Eq. (B.5). We found that the well-known least-squares method performs just as well as some modern algorithms used for this task, such as MINRES and its variants [51].

Finally, when the solution is found, we keep only the update function, $\mathbf{f}_{\text{geo}}$, discarding the Lagrange multipliers $\lambda$.

## C    Taming

Taming is a numerical procedure used in explicit integration of differential equations, particularly useful in preventing nonlinearity of equations from causing instabilities [30]. Its primary purpose is to control the magnitude of the gradient of the equation variables, in situations where the large gradient drives the evolution too far away from the correct trajectory. For a differential equation $\dot{\mathbf{y}} = \mathbf{f}(\mathbf{y}, t)$, the gradient is replaces according to the following rule:

$$\mathbf{f} \to \frac{\mathbf{f}}{1 + \mathrm{d}t \, \|\mathbf{f}\|}, \tag{C.1}$$

where $\mathrm{d}t$ is the numerical integration step. The taming procedure can be directly applied to the TDVP update function Eq. (5) in any formulation. We used taming throughout this work, as part of the attempts to control the breakdown for $\Delta = -2$ quench. However, we haven't noticed a benefit of taming in resolving this instability.

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
