# Peer review of "Instability of explicit time integration for strongly quenched dynamics with neural quantum states"

_SciPost Physics_

## Round 1 · Referee Report · Anonymous (Referee 1) · 2025-9-4

Report

The manuscript by Vrcan and Mentinik studies an interesting problem that has captured the attention of some researchers in the field of Neural Quantum States.

Their main message is that the equations of TDVP used to integrate a quench using variational quantum state encoded by an RBM can become 'unstable' when integrated with explicit numerical schemes, and that the only way to cure this unstability was to use implicit integrators.

In general, the thesis of the authors is interesting (and reasonable), but the numerical evidence reported is not enough in my opinion and the authors leave several questions unanswered. I think that some additional work would be warranted before publication. Assuming the authors properly clarify the unclear parts of their manuscript I see no holdup to publication.

The authors argue that the unstability is not emerging from a stiffness of the equations ('We also noticed no drastically varying time scales of dynamics, indicating no stiffness, which could otherwise cause numerical instabilities.'). However, the plots they report look like a textbook case of 'ODE stiffness', and i could not find any report in numerical analysis textbooks of other kind of 'numerical issues' that can lead to the issues discussed in the manuscript. 0. (All that follows assume that nothing is sampled, and quantities are computed exactly by summing over the full hilbert space) i. While i trust the analysis of the authors, I would appreciate if they could report numerical evidence (a figure) where they rule out stiffness of the equations. This can be done for example by computing the time-step error for RK45 and the 'target' timestep of the adaptive integrator. ii. I think the authors must do a better job to understand where the numerical unstability is coming from.

The problem the authors discuss makes me think of an issue that is faced by algorithms for multiscale physics, where for example one might want to simulate 'local' chemical reactions  at the same time as continental-scale atmospheric flows.
There, a major problem is that if you don't carefully chose the units, you end up with underflows of the effect of the 'local chemical reactions' (which contribute very little at each point). As far as i know (i'm not an expert) the solutions are a cmbination of carefully picking units such that velocities and forces are all within double-precision (15 ULPs) of eachother, hand-optimized code to avoid losing precision and summing different terms in very precise orders.

The reason this resonates is because RBMs are known to sometimes express some particular states setting some weights to very large numbers (>> 5 or << -5) while the others remain ~0, which when exponentiating could lead to a loss of precision. It would not be unreasonable to think that explicit integration of TDVP equations ends up representing the states where it breaks down with weights that have very different orders of magnitude, while implicit methods do not.

However, as the authors integrate the TDVP equations, i would expect that both explicit and implicit methods yield the same weights. 
So maybe it's just that implicit integrators are more resilient to loss of precision?
If they did their calculations in arbitrary precsion, would all methods yield the same results or not?

Regardless, I think the authors should identify a numerical experiment to see whther this is the problem and clearly prove what is causing it.

iii. If the problem is numerical in nature, it seems to me that it is strictly tied to the variational ansatz picked. For example, taking an exponential family like a Jastrow is usually stable and i would expect it to be stable also for those calculations. Likewise, a Vision Transformer, even if small and with only 1/2 layers, will lead to drastically different TDVP equations. Are those subject to unstabilities as well? Can the authors test with a small transformer with ~5-10k parameters if that's the case?

this point would be important to explain whether their results are generally relevant to the field or not much. In particular, if their result is not general to the architecture, the title of the manuscript should be changed.

iv. [this point is just a personal reflection...] The authors treat the TDVP equations as if they were an ODE that they can integrate with Heun/RK4 schemes. However the fact that the Fisher/S matrix is not invertible means that the TDVP equations are NOT a set of ODEs, but instead a particular kind of Differential Algebraic Equation (DAE), Mass Matrix DAEs.

In this domain, the null space of the S matrix would be seen as a set of constraints on the parameters, and you are 'obliged' to use implicit solvers because explicit ones would immediately violate the constraints after a timestep.

From this point of view, the result of the authors makes sense: they are seeing what mathematicians have long known, which is that to integrate DAEs you need implicit solvers. Now, this raises my next question: why does this only show up in particular points of the dynamics? Why were explicit integrators enough?

This is connected to my previous point: if it's a numerical precision issue, then doing calculations in arbitrary precision would mean that explicit solvers would also work. 
Instead, if it's not a numerical precision issue, but something deeper connectd to the structure of the DAE it could be identified by looking at the constraint violation error, in which case explicit methods would lead to a diverging constraint error near those points where there is a breakdown.

All in all, i'm asking the authors at the minimum to understand what is breaking down because of numerical issues before the manuscript can be accepted.

Recommendation

Ask for major revision

---

## Round 2 · Author Response

Dear Editor,

We hereby resubmit the manuscript for our paper, titled “Instability of explicit time integration for strongly quenched dynamics with neural quantum states.” The referee finds our results interesting and reasonable, but requests more numerical evidence, which we provide in the resubmitted manuscript. In particular, addressing the referee comments helped us identify that the nature of the instability can be mathematically understood as a stiff problem for the variational parameters, despite the fact that the exact quantum dynamics is not stiff. Evidence for this interpretation is provided by an analysis using an adaptive integrator. This finding is now included in the main text, including a new figure. We added several paragraphs in the Discussion section commenting on other possible origins of the numerical instability unmentioned in the original submission, as well as several appendices with figures supporting the analysis presented. Importantly, our main conclusion that alternative methods need to be developed to leverage the computational efficiency of explicit time integration of the TDVP equations remains still valid. We thank the referee for the constructive comments, which added valuable insights to our paper. We are confident that these changes will make the manuscript adequate for publication.

Please find our detailed answer to the referee comments and the list of changes on the submission page.

Sincerely,
Hrvoje Vrcan

---

## Round 2 · List of Changes

1. Added a sentence to the Abstract: “We provide evidence that the nature of the instability stems from stiffness of the dynamics of the variational parameters, despite the absence of stiffness in the exact quantum dynamics.”
  2. Added one sentence to paragraph 2 of the Introduction: “For example, in contrast to solving a continuous-time ODE of variational parameters, methods introduced in [10,13,14,22,24] solve an optimization problem at each step.” Rearranged some of the following sentences.
  3. Changes to the Discussion section: a. Paragraph 3: i. added a sentence with a reference to Appendix D, ii. removed a sentence about adaptive integration, iii. removed a sentence about stiffness. b. Added a new paragraph 4 about the stiffness of the equations of motion and the RK45 adaptive integrator. Other paragraphs have been shifted forward. c. Added a new Figure 5, with the results of the RK45 adaptive integrator. d. Added a sentence at the end of paragraph 5 about other neural network models. e. Added a new paragraph 6 about different ways to calculate quantum expectation values, and a previously reported biasing problem. f. Added a new paragraph 8, commenting on Differential Algebraic Equations (DAE-s).
  4. Changed several sentences in the Conclusion to address stiffness and the adaptive integrator.
  5. Added one sentence at the end of the Acknowledgements.
  6. Added Appendix D: Comparison of variational parameters a. Includes new Figure 6.
  7. Added Appendix E: RK45 adaptive integrator.
  8. Added Appendix F: Other neural network architectures. a. Includes new Figure 7.
  9. Added Appendix G: psi-logpsi formulations a. Includes new Figure 8.

---

## Editorial Decision

refereeing_in_preparation